# Similar Adaptations to 10 Weeks Concurrent Training on Metabolic Markers and Physical Performance in Young, Adult, and Older Adult Women

**DOI:** 10.3390/jcm10235582

**Published:** 2021-11-27

**Authors:** David C. Andrade, Marcelo Flores-Opazo, Luis Peñailillo, Pedro Delgado-Floody, Johnattan Cano-Montoya, Jaime A. Vásquez-Gómez, Cristian Alvarez

**Affiliations:** 1Centro de Investigación en Fisiología y Medicina de Altura (FiMedAlt), Departamento Biomédico, Facultad de Ciencias de la Salud, Universidad de Antofagasta #02800, Antofagasta 1271155, Chile; david.andrade@uantof.cl; 2Laboratory of Exercise Physiology and Metabolism (LABFEM), Department of Physiotherapy, Faculty of Medicine, Universidad Finis Terrae, Santiago 7501015, Chile; mflores@uft.cl; 3Exercise and Rehabilitation Sciences Laboratory, School of Physical Therapy, Faculty of Rehabilitation Sciences, Universidad Andres Bello, Santiago 7591538, Chile; luis.penailillo@unab.cl; 4Department of Physical Education, Sport and Recreation, Universidad de La Frontera, Temuco 4780000, Chile; pedro.delgado@ufrontera.cl; 5Escuela de Kinesiología, Facultad de Ciencias de la Salud, Universidad San Sebastián, Valdivia 8420524, Chile; jcanom@docente.uss.cl; 6Centro de Investigación de Estudios Avanzados del Maule (CIEAM), Universidad Católica del Maule, Talca 3460000, Chile; jvasquez@ucm.cl; 7Laboratorio de Rendimiento Humano, Grupo de Estudios en Educación, Actividad Física y Salud (GEEAFyS), Universidad Católica del Maule, Talca 3460000, Chile; 8Quality of Life and Wellness Research Group, Department of Health, Universidad de Los Lagos, Osorno 5290000, Chile

**Keywords:** concurrent training, risk factors, metabolic syndrome, hypertension, type 2 diabetes mellitus, adults, women

## Abstract

It has been proposed that the combination of high-intensity aerobic exercises and resistance training (RT) known as concurrent training (CT) could improve metabolic syndrome (MetS) markers, and that the exercise mixture in CT could dampen muscle anaerobic pathways, a result known as the interference effect. However, there is scarce evidence on its effects in women across different ages. Therefore, we sought to determine the effects of a 10-week CT intervention on MetS markers and endurance performance in adult women and compared age-related differences between young, adult, and older participants. A total of 112 women with >1 MetS risk factors were included in the study. Participants were allocated to different groups according to the following cutoff age ranges: 20–29years (y), *n* = 25; 30–39y, *n* = 35; 40–49y, *n* = 43; and 50–59y, *n* = 53. Participants performed 10 weeks of CT, including resistance training (RT), involving six major muscle groups, and high-intensity interval training (HIIT) in a cycle ergometer. Anthropometric, cardiovascular, metabolic, and performance outcomes were assessed before and after the intervention. The CT induced significant improvements in waist circumference (WC) (20–29y: –2.5; 30–39y: –4.1; 40–49y: –4.2; 50–59y: –2.8 Δcm) and the distance achieved in the six-minute walking test (6Mwt) (20–29y: +47.6; 30–39y: +66.0; 40–49y: +43.0; 50–59y: +58.6 Δm) across all age groups, without significant differences between groups. In addition, a significant correlation was found between 6Mwt and WC, independent of age. In conclusion, our results showed that a 10-week CT intervention improved MetS risk factors in women, suggesting that the beneficial effects promoted by CT are independent of age and confirming CT as an effective, age-independent training regimen to improve metabolic health in women.

## 1. Introduction

The modern lifestyle in urbanized societies has become a burden on health and wellbeing, owing to its association with unhealthy lifestyle habits characterized by calorie-dense diets and physical inactivity [1]. In particular, the adult population has seen increases in their cardiometabolic risk for type 2 diabetes (T2DM) [2] and hypertension (HTN) [3]. As a result, women are more physically inactive (23.1 vs. 17.1%), and have a higher prevalence of T2DM, than men (10.4 vs. 8.4%) [4]. This suggests that physically inactive women are more susceptible to metabolic disorders or metabolic syndrome (MetS). MetS is a cluster of cardio-metabolic alterations—including central obesity, dyslipidemia, high blood pressure (BP), and hyperglycemia—which increases the risk for T2DM, HTN, and mortality [5].

Exercise training has proven to be an effective non-pharmacological therapy on T2DM, HTN, and MetS risk factor management [6,7,8,9,10]. Exercise training reduces body fat and improves metabolic control, cardiovascular functions, and exerts anti-inflammatory effects [6,9,11,12]. Although exercise training is able to promote the maintenance of health and prevent MetS [9], the evidence suggests that the exercise-induced reduction in the cardiovascular disease risk load could be decreased with age [13,14].

The combination of aerobic and resistance exercise modalities, or so-called concurrent training (CT), has been shown to improve vascular functionality [15,16,17,18], lipid profile, cardiorespiratory fitness, and body composition. It has also been proven to reduce systolic blood pressure and fasting glycemia in sedentary adults with impaired cardio-metabolic health [6,8,19]. Notably, these effects are greater compared with either aerobic or resistance training alone [20,21]. Despite this, there is evidence that CT could compromise the adaptations induced by each modality alone [22,23]. Mechanistically, it has been stated that signaling responses mediating adaptations of endurance training involved in mitochondrial biogenesis (i.e., adenosine monophosphate (AMP)-activated protein kinase (AMPK), Ca^2+^/calmodulin-dependent kinase II (CaMKII), and peroxisome proliferator-activated receptor-c coactivator-1 (PGC-1a)) seem to dampen muscle anabolic pathways activated by resistance training (i.e., mechanistic target of rapamycin complex 1 (mTORC1) and downstream effectors 70 kDa ribosomal S6 protein kinase (S6K) and eukaryotic initiation factor 4E binding protein (4E-BP)) [23,24,25]. This effect has been denominated the interference effect [26], but it has not been made clear whether or not this phenomenon is activated in untrained or trained women as well as across different ages [26,27,28,29].

Although the molecular mechanisms underlying exercise-induced metabolic and performance have been documented, the beneficial effects of exercise training on MetS across different ages have not been very well described. Hence, it was deemed necessary to determine whether CT could promote comparable adaptation across different ages in women. This study aimed to determine the effects of 10 weeks of CT on MetS risk factors across young-to-older women, and their association with endurance performance. We hypothesized the presence of similar adaptations for improving MetS outcomes among young-to-older women.

## 2. Materials and Methods

The present study was a quasi-experimental (non-randomized), clinical, and multicenter intervention, carried out between 2015 and 2018 in Los Lagos (2015–2016) and Rio Bueno (2017–2018) in Chile. The study included participants from urban and rural communities (including Chilean ascendants from European and ethnic groups) in both cities, were recruited from two community health centers: (1) Family Healthcare Centre Tomás Rojas of Los Lagos and (2) the primary health center, General Urban Healthcare Centre, in Río Bueno city. Recruitment was advertised publicly through local media and participants from each area were directly asked to take part. All participants read a detailed description of the study protocol and provided their written informed consent. The study was part of Clinicaltrials.gov (ID NCT03653468) and was conducted according to the Declaration of Helsinki. All interventions were approved by the ethics committee of the Family Healthcare Centre Tomás Rojas, Chile in 2016.

The eligibility criteria required participants: (i) to be women experiencing physical inactivity as measured by the International Physical Activity Questionnaire (IPAQ) applied previously in Chile (<600 metabolic equivalents (MET)·min/week [30]), (ii) to have fasting hyperglycemia > 100 mg/dL and <126 mg/dL [31], and (iii) to have other additional MetS markers, such as 1+ values above normal cutoffs for body composition (i.e., WC > 80 cm, which indicates high cardiovascular risk in South American populations [32]); systolic (SBP) or diastolic (DBP) blood pressure of 130–139 mmHg and 85–90 mmHg, respectively, denoting high blood pressure, SBP and DBP > 140 mmHg and 90 mmHg, respectively, denoting hypertension [33], or lipid profile (i.e., total cholesterol > 200 mg/dL) [32]. Exclusion criteria were as follows; (i) cardiovascular contraindications to exercise, (ii) history of stroke, asthma, or chronic obstructive pulmonary disease, (iii) muscle-skeletal disorders, (iv) smoking, (v) living in extremely rural areas, (vi) enrolled in other exercise training programs, or (vii) T2DM or HTN diagnosis. A compliance rate to the exercise program ≥ 70% was required for the participants in the intervention group to be included in the statistical analyses.

The sample size was calculated using delta change observed in fasting plasma glucose (∆FPG = 2.3 ± 1.7 mg/dL, mean ± standard deviation [SD]) from a previous study involving individuals of the same age range that underwent a similar intervention [34]. A statistical power analysis revealed that a total of 12 participants per group would yield a power of 80% at a *p* < 0.05 alpha level. A total of (*n* = 232) participants were recruited according to the eligibility criteria. Fifty-eight subjects were excluded at the initial stage and forty-six withdrew at some point throughout the study. The remaining (*n* = 174) participants were divided into four groups according to age: 20–29, 30–39, 40–49, and 50–59 years. Thus, the final sample was as follows in mean (95% CI): (a) 20–29y group [*n* = 28, age 25.2 (23.7, 26.6) years], (b) 30–39y group [*n* = 35, age 35.4 (34.4, 36.3) years], (c) 40–49y group [*n* = 28, age 42.8 (41.3, 44.3) years], and (d) 50–59y group [*n* = 27, age 53.1 (50.2, 56.0) years]. The study design is shown in (Figure 1).

### 2.1. Anthropometric Measurements

The following anthropometry endpoints were measured before and after the training intervention in all groups: body mass (kg), body mass index (kg/m^2^), and waist circumference. Measurements were taken in the morning between 8:00 and 11:00 a.m. while participants were barefoot and nonmetal devices were worn during the process. Body mass (in kg, to the nearest 0.1 kg) and height (in m, to the nearest 0.1 m precision) were assessed by a professional scale (Health o Meter^®^ Professional, Sunbeam Products Inc., Chicago, IL, USA). Body mass index (BMI) was calculated as the body mass divided by the square of the body height. Waist circumference (WC) in cm was measured using a flexible and inextensible measuring tape (Hoechstmas^®^, Sulzbach, Germany).

### 2.2. Blood Pressure Measurement

Systolic and diastolic blood pressure (SBP and DBP, respectively) were assessed at baseline and at the last session of the 10–week CT intervention. From SBP and DBP we calculated mean arterial pressure (MAP) (1/3 of SBP + 1/2 of DBP). Blood pressure was determined using an automated blood pressure monitor (Omron^®^ HEM 7114, Omron Healthcare Inc., Lake Forest, IL, USA). Readings were taken from the left arm, in triplicate (2-min interval between measurements) after 15 min of rest with the subjects in a seated position.

### 2.3. Fasting Plasma Glucose and Lipid Profile Measurement

Participants arrived at the healthcare center between 8:00 and 10:00 a.m. following a 12–h fasting period. Blood samples were drained from the antecubital vein and collected in serum and plasma tubes (BD Vacutiner^®^, Becton Dickinson Company, Franklin Lakes, New Jersey, USA). Samples were immediately placed on ice and centrifuged at 4000 rpm for 5 min, after which plasma and serum fractions were aliquoted and immediately stored at –20 °C until analysis. Fasting plasma glucose (FPG) was analyzed using an enzymatic glucose oxidase assay (GOD-PAP; Wiener Lab, enzyme, Rosario, Santa Fe, Argentina). Plasma total cholesterol (TC), and triglycerides (Tg) were analyzed using standard methods with an automatized analyzer (Metrolab Biomed^®^, Model 2300 PLUS, Buenos Aires, Argentina). HDL-c levels were determined using standard enzymatic methods with ad hoc reagents, LDL-c and VLDL-c levels were calculated using the Friedewald formula [35], and the TC/HDL-c ratio was determined accordingly.

### 2.4. Endurance Performance Measurement

The endurance performance was assessed using the six-minute walk test (6Mwt). The test was carried out in the morning between 8:00 and 12:00 a.m. on an indoor sports court. A 10-min warm-up including low-intensity walking and full range–of–motion movements in knees and ankles was performed before the test. Participants were then instructed to walk as fast as possible according to their individual capacity in a 100-m track for six minutes or until participants manifested stopping. The total distance walked was recorded and registered in meters, and their validation and protocol have been widely described and used previously by the National Health Chilean Survey [36].

### 2.5. Concurrent Exercise Training Program

Before the intervention, all subjects performed three sessions of familiarization of CT, and afterward, the program was developed into 3 days/week (2 sessions guided by a professional, and 1 session self-guided, with previous education before starting the program). Each CT training session was divided into two parts: (1) resistance training (RT), where participants completed free weights RT exercises aimed at six major muscle groups (i.e., biceps curl, shoulder press, upper row, squat), and (2) high-intensity interval training (HIIT) in a cycle ergometer (BH^®^, model Carbon Bike Generator, Santiago, Chile). The resistance training modality consisted of three series of 1:1 min work to rest ratios. On each series, the number of repetitions was maintained until fatigue, defined as reaching a rate of perceived exertion (RPE) of 8 on the 0–10 modified Borg scale [37]. The RT intensity corresponded to 20–40% of one-repetition maximum test (1RM) for each exercise at the start of the study and progressed to a 25–50% of 1RM at the study end. The HIIT exercise modality included 60-s bouts of pedaling at 8–10/10 RPE, followed by 120 s of passive recovery on the ergometer, as previously described [34]. The number of bouts increased during the program, beginning at 3 between weeks 1 and 4 and progressing to 5–7 between weeks 5 and 7 and 8–10 between weeks 8 and 10. The total length of the CT session was in the mean of 54 min (RT; 24 min and HIIT; 12 to 30 min). The total number of CT sessions were 30 (100% adherence = 30 sessions, wherein 20 sessions were supervised and the other 10 were self-guided). The minimum adherence expected was 70% guided sessions (i.e., 14 sessions), and at least 50% adherence for the self-guided sessions (i.e., 5 sessions).

### 2.6. Statistical Analysis

The data in the table is shown as mean and 95% confidence intervals (CIs) and as mean and (±) standard error of the mean (SEM) in figures. Normality and homoscedasticity assumptions for all data were determined by Shapiro–Wilk and Levene (*F*) tests, respectively. Univariant analysis of variance (ANOVA) was conducted to identify differences between groups at baseline. After the intervention, delta changes (∆) were calculated for each outcome and comparisons between groups were performed with a one-way ANOVA, followed by a Sidak’s post hoc test for multiple comparisons. Eta partial squared for interaction (Time × Group) was assessed by *η*^2^ obtained from the analysis of covariance (ANCOVA) with small (*η*^2^ = 0.01), medium (*η*^2^ = 0.06), and large (*η*^2^ = 0.14) effects defined according to Lakens [38]. The level of significance was set at *p* < 0.05. The study data were processed using SPSS for Windows, program version 23.0 (SPSS^®^ Inc., Chicago, IL, USA), and graphs were created using GraphPad Prism 8.0 software (GraphPad Software, San Diego, CA, USA).

## 3. Results

### 3.1. Baseline Characteristics

Baseline characteristics are summarized in the Table 1. No significant baseline differences were found between age groups in primary MetS outcomes such as WC [*F*_(0.70)_, *p* = 0.591], SBP [*F*_(0.17)_, *p* = 0.949], DBP [*F*_(1.27)_, *p* = 0.286], HDL-c [*F*_(1.55)_, *p* = 0.196], Tg [*F*_(2.37)_, *p* = 0.060], and FPG [*F*_(1.07)_, *p* = 0.374], or in secondary outcomes such as body mass [*F*_(3.13)_, *p* = 0.081], BMI [*F*_(0.26)_, *p* = 0.901], TC [*F*_(0.98)_, *p* = 0.324], LDL-c [*F*_(1.97)_, *p* = 0.106], VLDL-c [*F*_(2.37)_, *p* = 0.060], TC/HDL-c Ratio [*F*_(1.05)_, *p* = 0.384], MAP [*F*_(0.88)_, *p* = 0.480], and 6Mwt at baseline [*F*_(0.28)_, *p* = 0.890] between groups (Table 1).

### 3.2. Training-Induce Effects

The effects of CT intervention on primary and secondary outcomes are shown in Table 1. Our data revealed that the groups differed in the number of primary outcomes in MetS markers promoted by CT intervention. Indeed, the 20–29y group decreased (*p* < 0.05) three (WC, *p* < 0.0001; SBP, *p* = 0.011; and FPG, *p* = 0.013), the 30–39y only one (WC, *p* < 0.0001), the 40–49y group three (WC, *p* < 0.0001; Tg, *p* = 0.014; and FPG, *p* = 0.048), and the 50–59 age group only one (WC, *p* = 0.002) MetS outcomes after CT (Table 1).

Similarly, our results showed that the effects of CT intervention on secondary outcomes also differed between groups. Indeed, in the 20–29y group, CT promoted significant training-induced changes in body mass (*p* < 0.0001), BMI (*p* < 0.0001), and 6Mwt (*p* < 0.0001) (Table 1). In the 30–39y group, there were significant changes in BMI (*p* = 0.023) and 6Mwt (*p* < 0.0001) (Table 1). In the 40–49y group, there were significant changes in body mass (*p* < 0.0001), BMI (*p* < 0.0001), TC (*p* = 0.029), LDL-c (*p* = 0.013), VLDL-c (*p* = 0.014), ratio TC/HDL-c (*p* = 0.024), and 6Mwt (*p* < 0.0001). In the 50–59y group, there were significant changes in body mass (*p* = 0.004), and 6Mwt (*p* = 0.037) (Table 1).

### 3.3. Age-Group Comparisons at Main MetS Outcomes

Similar changes on primary outcomes were found between age groups (Figure 2). Of note, WC (∆WC *F*_(0.17)_, *p* = 0.910, *η*^2^ = 0.007) (Figure 2a), FPG (∆FPG *F*_(0.50)_, *p* = 0.677, *η*^2^ = 0.02) (Figure 2b), SBP (∆SBP *F*_(0.79)_, *p* = 0.156, *η*^2^ = 0.07) (Figure 2c), DBP (∆DBP *F*_(0.30)_, *p* = 0.820, *η*^2^ = 0.01) (Figure 2d), HDL-c (∆HDL-c *F*_(0.85)_, *p* = 0.469, *η*^2^ = 0.03) (Figure 2e), and Tg (∆Tg *F*_(1.89)_, *p* = 0.139, *η*^2^ = 0.07) (Figure 2f) did not differ between groups, suggesting that the effects promoted by the exercise intervention were similar between different age groups.

Similarly to primary outcomes, the ∆ effects promoted by CT intervention on secondary outcomes did not differ between groups (Figure 3). Specifically, body mass (∆body mass *F*_(0.79)_, *p* = 0.503, *η*^2^ = 0.03) (Figure 3a), BMI (∆BMI *F*_(1.05)_, *p* = 0.372, *η*^2^ = 0.04) (Figure 3b), MAP (∆MAP *F*_(0.39)_, *p* = 0.754, *η*^2^ = 0.01) (Figure 3c), LDL-c (∆LDL-c *F*_(2.88)_, *p* = 0.042, *η*^2^ = 0.10) (Figure 3e), VLDL-c (∆VLDL-c *F*_(1.89)_, *p* = 0.139, *η*^2^ = 0.07) (Figure 3f) and 6Mwt (∆6Mwt *F*_(1.80)_, *p* = 0.154, *η*^2^ = 0.07) (Figure 3h) were similar between groups. However, we found group-related differences (ANOVA) in TC (∆TC *F*_(3.13)_, *p* = 0.031, *η*^2^ = 0.11) (Figure 3d) and LDL-c (∆LDL-c *F*_(2.88)_, *p* = 0.042, *η*^2^ = 0.10) (Figure 3e). Nevertheless, pairs comparisons did not reveal significant differences between age-groups. All of these results suggested that the effects promoted by the exercise were independent of age.

A correlation analysis between ∆6Mwt, and primary and secondary MetS outcomes was conducted to determine whether an association existed between the improvement in 6Mwt and the CT-induced effects on these variables independently of age (Table 2). A significant association was found between ∆6Mwt and ∆WC (r = −0.26, *p =* 0.017) independently of age (Table 2). A trend toward a negative association was found between improvements in 6Mwt and body weight loss (r = −0.21, *p =* 0.054). The rest of the MetS outcomes were not modified by changes in 6Mwt.

## 4. Discussion

The purpose of the present study was to determine the effects of 10-weeks of CT on MetS risk factors across young-to-older women and their association with endurance performance. Our results showed that: (i) the 10-week CT program decreased adiposity (i.e., WC) as well as FPG, SBP/DBP, HDL-c, and Tg similarly across youn-to-older women at risk of MetS; (ii) the beneficial effects promoted by CT intervention were similar across all groups; and (iii) ∆6Mwt improvement was significantly correlated with ∆WC, independent of age. Thus, our data strongly suggested that CT intervention is an effective non-pharmacological strategy to improve MetS outcomes independent of age. Therefore, our hypothesis was supported by our results, showing similar beneficial adaptations induced by CT in all age groups.

Exercise is considered a first line intervention to prevent and/or treat MetS in adult populations. Accordingly, moderate-intensity continuous training (MICT) has been shown to induce strong health benefits, including decreases in TC, Tg, and BP, and increases in HDL-c, peripheral and central vascular functionality, and aerobic capacity [11,12,39,40]. Although the beneficial effects of exercise are very well accepted, results are not consistent across studies. Dumortier et al. [39] showed that an 8-week MICT (3 times × wk, at intensity, targeted to obtain maximal fat oxidation) improved body composition parameters (∆Body mass −2.6 kg, ∆BMI −1.0 kg/m^2^, ∆WC −4.2 cm), but was ineffective for improving lipid profile and FPG in middle-age MetS adult patients [39]. Similarly, Green et al. [40], analyzing data from the HERITAGE study, demonstrated that 20-weeks of MICT (30–10 min/session, at 55–15% of the VO_2max_) did not induce changes in body composition, blood pressure, and FPG, but induced improvements in lipid profile (∆HDL-c +2.5 mg/dL, ∆Tg −3.6 mg/dL) and cardiorespiratory fitness (∆VO_2max_: +4.3 mL/kg/min) [40]. Alternatively, HIIT has proven to render superior improvements in glycemic control and insulin sensitivity, blood pressure, skeletal muscle oxidative capacity, and lipid profile in individuals at risk of cardiometabolic diseases [6,7,8,41]. Thus, endurance training, either continuous or in intervals, is a helpful strategy to aid in reversing metabolic alterations associated with MetS. On the other hand, RT has been reported to induce considerable beneficial effects on adiposity, blood pressure, lipid metabolism, and glycemic control in individuals of different ages and health statuses [42,43]. Indeed, Castaneda et al. [42] reported that 16-weeks of RT (45 min/session, 60–10% of 1RM, 3 times/week) decreased systolic BP (∆SBP −10 mmHg), FPG (∆FPG −11 mmol/L), and Tg (∆Tg −13.8%) in women over 65 years of age. In this line, we have previously reported improvements in adiposity (∆WC −3.0 cm), blood pressure (∆SBP −4 mmHg) FPG (∆FPG −6 mg/dL), and endurance performance (∆ −2 min in 2 km walking test) following 12-weeks of RT in sedentary overweight/obese insulin-resistant young women [34].

In the present study, we found that 10-week CT intervention, including both RT and HIIT, promoted positive training-induced effects on MetS markers, such as central adiposity, blood pressure, lipid profile, and fasting glucose in all age groups, suggesting that this type of training could be effective in improving MetS factors across age groups in women at risk of this syndrome. Importantly, we found that both primary and secondary outcomes could be positively (but differently) affected by CT intervention, suggesting that our results could be relevant for clinical interventions in this population. Thus, as our data revealed robust effects across all ages, it is possible to summarily state that the CT program could promote beneficial effects independently of age. However, TC (Figure 3d) and LDL-c (Figure 3e) displayed a significant interaction between groups. Nevertheless, our pair comparison analyses did not reveal significant differences between all groups. Of note, the older 50–19y group were more sensitive to improvements in secondary outcomes such as TC (∆ −12.8 mg/dL) and LDL-c (∆ −8.9 mg/dL) following CT intervention, which suggests that this population could get more health benefits from CT. However, as our study did not detect significant differences across ages, future studies need to address this important point, increasing the number of participants and/or applying different CT strategies in this particular population.

Concurrent training is an exercise modality from which participants can benefit as it induces both cardiometabolic and physical fitness adaptations, an aspect of importance when prescribing exercise to older individuals. Recently, we reported muscle mass and strength gains, concomitantly with improvements in body composition (fat mass, WC), FPG, blood pressure, and circulating lipid levels in overweight women at high risk of cardiometabolic alterations followed CT intervention [6,8]. Similarly, Atashak et al. [19] reported improvements in lipid profile and soluble intercellular adhesion molecule-1 (sICAM-1), a circulating marker of inflammation highly correlated with high blood pressure in middle-aged (30–10y) men after an 8-week (3 times per week) progressive CT [19]. Our 10-week CT intervention, combining RT and HIIT, had similar adaptations, independent of age, in line with previous reports [44]. It is difficult to compare our CT program (~54 min/session of RT and HIIT) with other more time-efficient regimes, such as HIIT, that promote specific benefits to cardiovascular and metabolic health [44]. However, by contrast, comparing our CT with 1-year of endurance training [45], it is relevant to highlight that 12-weeks of CT induced integrally relevant benefits to body mass, lipid profile, and endurance performance. Likewise, CT versus long-time endurance training appears more volume efficient, promoting more integral health benefits in physically inactive adults at risk of MetS. Thus, findings from the present study may have important health repercussions in favor of the prescription of CT to decrease the MetS risk factors in physically inactive adults.

Some studies have suggested that a differential improvement in cardio-respiratory fitness (CRF) could cause divergent responses in reducing MetS severity [46,47]. Earnest et al. [48] suggested that lower initial insulin sensitivity was associated with a greater response to any dose of exercise [48]. Accordingly, in our study, neither CRF (assessed by 6Mwt) nor glycemic control (determined by FPG) differed between groups, considering that all subjects displayed similar values at 6Mwt and FPG before the CT intervention. Thus, we speculated that factors such as adherence, progression, and adaptation to the intensity/recovery periods may have influenced results to a greater extent than chronological age itself, which could have generated different adaptations between distinct chronological ages. However, previous studies have shown that age is not a relevant factor in promotion of CRF adaptations to exercise training [49]. Our study also provided some clinical relevance; for example, a reduction in cardiometabolic risk factors can positively modify disease mortality and healthcare expenditures [50]. Decreases of ~0.9 to 2.0 mmHg on SBP or DBP reduced major cardiovascular events by 10% in T2DM patients [51], and HDL-c increases (~10 mg/dL) can lead to other clinical beneficial implications. Thus, as we reported SBP decreases of 0.1−1.6 mmHg and other HDL-c increases from +0.4 to +3.1 mg/dL, our study provides relevant clinical implications to physically inactive cohorts at risk of MetS.

### Strengths and Limitations

The present study was not free of limitations. Some limitations of this study included the fact that (i) daily life activity was not measured across the four age groups; thus, it is possible that daily physical activity differed between groups, which could have influenced the results after training. Additionally, (ii) we did not control additional physical activity (i.e., sports participation) and diet, however, we regularly (i.e., each week) asked the participants to maintain their initial physical activity/exercise level. Moreover, (iii) we did not include a control group—although the young exercise group could be considered to serve this purpose—and (iv) we included participants from urban and rural communities (i.e., including Chilean ascendants from European and ethnic Amerindian groups), which could lead to potential bias in the data interpretation, (v) we did not control for menstrual cycle or menopausal symptoms, we did not register reports of participants during the follow-up about this concern, and (vi) our results were not controlled for reproductive hormone fluctuations. However, judging by our previous experience in physically inactive women groups, we did not report reduction of beneficial effects. Some strengths of this study include the fact that (i) in addition to MetS outcomes, we included other anthropometric, cardiovascular, metabolic, and endurance performance outcomes, and finally, (ii) we reported a Latin-American female sample of workers/students of different ages groups in response to CT.

## 5. Conclusions

In conclusion, findings in the present study suggested a similar adaptation to CT, which was found across a large range of ages in women (20 to 60 years old). The main results showed significant improvements in central adiposity, FPG, and lipid profile. Thus, these findings should be considered clinically relevant, showing that age is not a limitation on the health benefits induced by exercise, and specifically CT setting interventions.

## Figures and Tables

**Figure 1 jcm-10-05582-f001:**
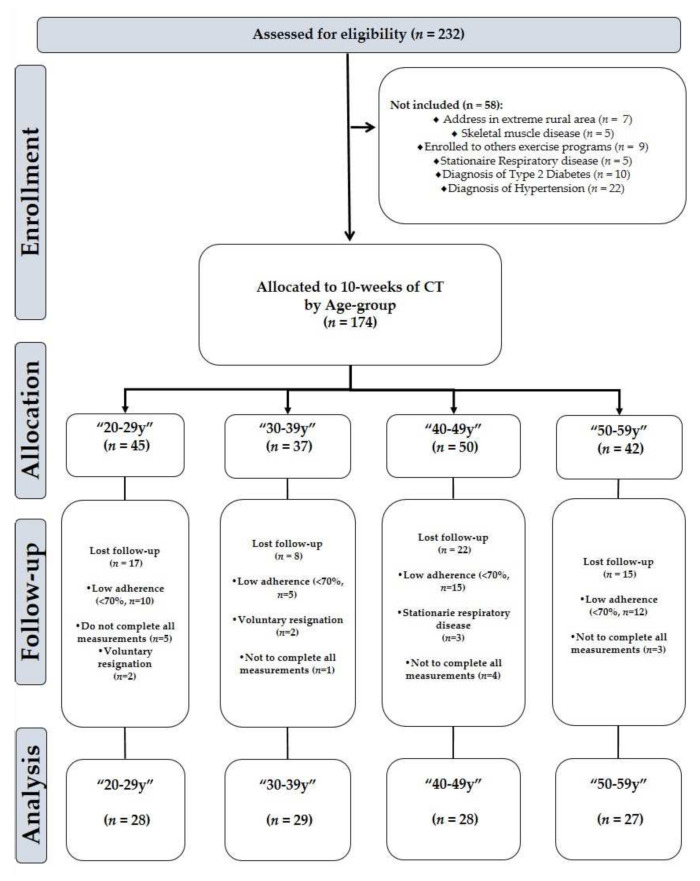
Flow Chart.

**Figure 2 jcm-10-05582-f002:**
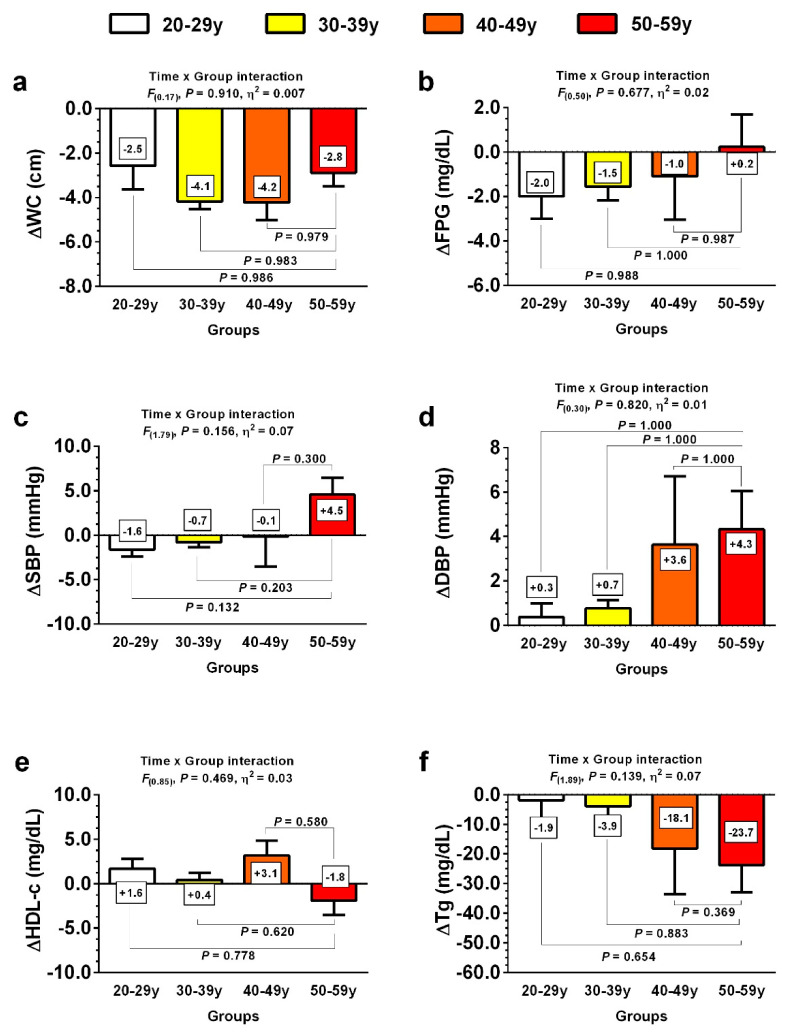
Changes in MetS primary outcomes following 10 weeks of concurrent training in four different age groups. Groups are described as: (20–29y), adults from 20 to 29 years, (30–39y), adults from 30 to 39 years, (40–49y), adults from 40 to 49 years, and (50–59y), adults from 50 to 59 years. Variables are described as: (**a**) ∆WC, Delta waist circumference; (**b**) ∆FPG, delta fasting plasma glucose; (**c**) ∆SBP, delta systolic blood pressure; (**d**) ∆DBP, delta diastolic blood pressure; (**e**) ∆HDL-c, delta high-density lipoprotein cholesterol; (**f**) ∆Tg, delta triglycerides; *F* denotes Levene test; *η*^2^ denotes Lakens effect size.

**Figure 3 jcm-10-05582-f003:**
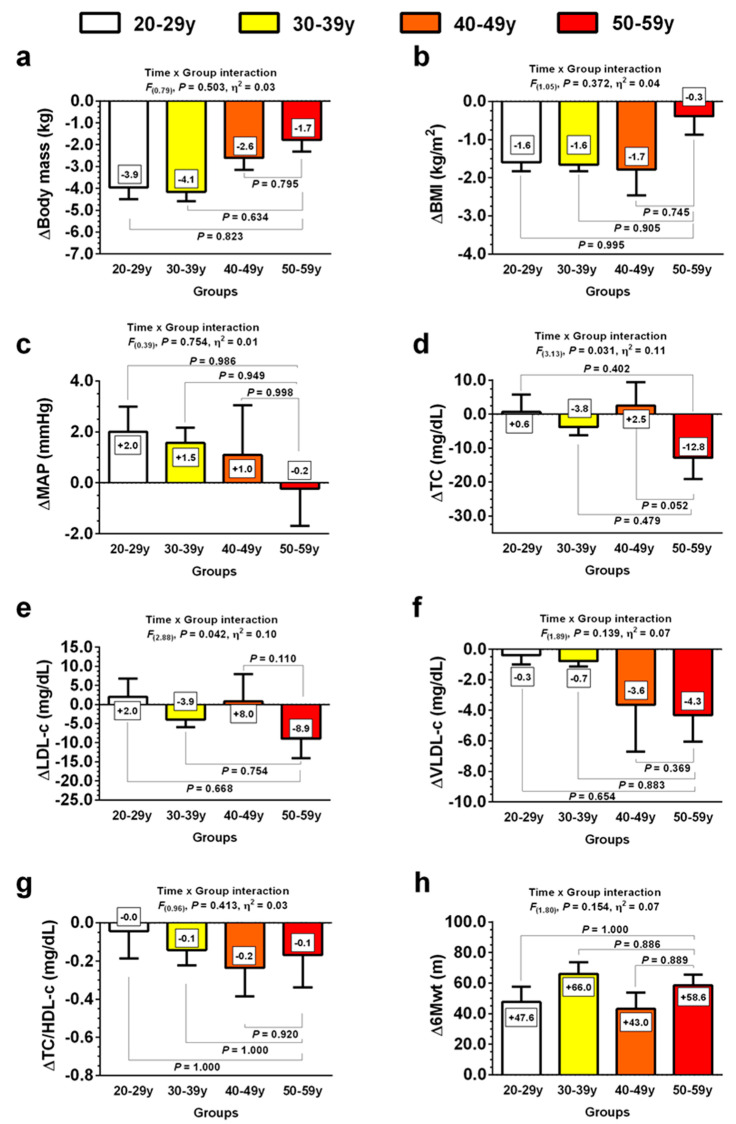
Changes in MetS secondary outcomes following 10 weeks of concurrent training in four different age groups. Groups are described as: (20–29y), adults from 20 to 29 years, (30–39y), adults from 30 to 39 years, (40–49y), adults from 40 to 49 years, and (50–59y), adults from 50 to 59 years. Variables are described as: (**a**) (∆Body mass), delta body mass; (**b**) (∆BMI), delta body mass; (**c**) (∆MAP), delta mean arterial pressure; (**d**) (∆TC), delta total cholesterol; (**e**) (∆LDL-c), delta low-density lipoprotein cholesterol; (**f**) (∆VLDL-c), delta very-low density lipoprotein cholesterol; (**g**) (∆TC/HDL-c), delta ratio total cholesterol/high-density lipoprotein cholesterol; (**h**) (∆6Mwt), delta 6 min walking test; *F* denotes Levene test; *η*^2^ denotes Lakens effect size.

**Table 1 jcm-10-05582-t001:** Baseline characteristics and effects of CT intervention in women with MetS risk factors.

MetS Outcomes	Groups				Baseline, *F* and *p* Value
	20–29y ^a^	30–39y ^b^	40–49y ^c^	50–59y ^d^	
*n* =	28	29	28	27	
Age (y)	25.2 (23.7; 26.6)	35.4 (34.4; 36.3)	42.8 (41.3; 44.3)	53.1 (50.2; 56.0) ^abc^	(150.2), ***p* < 0.0001**
**Primary outcomes**					
Waist circumference (cm) ∆pre-post (%) Pre-Post *p*-value	98.4 (92.7; 104.0) **−3.2** ***p* = 0.001**	98.0 (93.9; 102.1) **−3.5** ***p* < 0.0001**	102.0 (97.5; 106.5) **−3.5** ***p* < 0.0001**	101.5 (94.1; 108.9) **−4.2** ***p* = 0.002**	(0.70), *p* = 0.591
Systolic BP (mmHg) ∆pre-post (%) Pre-Post *p*-value	118.8 (113.3; 124.3) **−3.0** ***p* = 0.011**	116.4 (110.8; 122.0) −1.0 *p* = 0.424	119.2 (113.1; 125.3) −1.0 *p* = 0.269	117.6 (103.6; 131.6) 3.8 *p* = 0.552	(0.17), *p* = 0.949
Diastolic BP (mmHg) ∆pre-post (%) Pre-Post *p*-value	80.5 (75.5; 85.6) 0.3 *p* = 0.837	74.2 (68.7; 79.7) 6.9 *p* = 0.218	80.9 (76.1; 85.8) 4.4 *p* = 0.523	79.1 (71.8; 86.4) 5.4 *p* = 0.571	(1.27), *p* = 0.286
HDL-cholesterol (mg/dL) ∆pre-post (%) Pre-Post *p*-value	51.3 (46.6; 55.9) 1.1 *p* = 0.845	53.0 (48.9; 57.6) 1.1 *p* = 0.815	53.3 (47.0; 59.7) 1.5 *p* > 0.999	51.3 (45.0; 57.6) 6.9 *p* = 0.178	(1.55), *p* = 0.196
Triglycerides (mg/dL) ∆pre-post (%) Pre-Post *p*-value	108.3 (86.4; 130.3) −2.5 *p* = 0.137	104.0 (90.3; 117.7) −2.1 *p* = 0.096	150.5 (111.5; 189.4) **−10.6** ***p* = 0.014**	124.1 (87.1; 161.0) −7.0 *p* = 0.412	(2.37), *p* = 0.60
Fasting plasma glucose (mg/dL) ∆pre-post (%) Pre-Post *p*-value	93.8 (909.1; 97.5) **−2.0** ***p* = 0.013**	93.1 (90.4; 95.8) −0.03 *p* = 0.938	96.8 (91.5; 102.1) **−1.3** ***p* = 0.048**	97.6 (91.0; 104.2) −0.2 *p* = 0.962	(1.07), *p* = 0.374
**Secondary outcomes**					
Body mass (kg) ∆pre-post (%) Pre-Post *p*-value	79.8 (71.9; 87.7) **−5.3** ***p* < 0.0001**	78.2 (70.7; 85.8) −5.5 *p* = 0.084	77.4 (71.9; 82.8) **−7.2** ***p* < 0.0001**	72.8 (66.4; 79.2) **−3.5** ***p* = 0.004**	(3.13), *p* = 0.081
Body mass index (kg/m^2^) ∆pre-post (%) Pre-Post *p*-value	31.8 (28.1; 35.4) **−5.3** ***p* < 0.0001**	31.1 (28.2; 33.9) **−2.8** ***p* = 0.023**	31.3 (29.0; 33.5) **−3.0** ***p* < 0.0001**	30.7 (28.4; 33.0) −6.5 *p* = 0.058	(0.26), *p* = 0.901
Total cholesterol (mg/dL) ∆pre-post (%) Pre-Post *p*-value	186.0 (157.2; 214.7) −0.8 *p* = 0.359	173.7 (164.6; 182.7) −0.03 *p* = 0.752	199.0 (182.0; 216.0) **−6.0** ***p* = 0.029**	191.7 (159.1; 224.4) 4.1 *p* = 0.517	(0.98), *p* = 0.324
LDL-cholesterol (mg/dL) ∆pre-post (%) Pre-Post *p*-value	113.9 (89.4; 138.4) 1.5 *p* = 0.532	99.0 (90.9; 107.1) 1.2 *p* = 0.774	115.3 (102.5; 128.2) **−9.1** ***p* = 0.013**	115.5 (89.3; 141.7) 4.9 *p* = 0.769	(1.97), *p* = 0.106
VLDL-cholesterol (mg/dL) ∆pre-post (%) Pre-Post *p*-value	21.6 (17.2; 26.0) −2.5 *p* = 0.137	20.8 (18.0; 23.5) −2.1 *p* = 0.096	30.1 (22.3; 37.8) **−10.6** ***p* = 0.014**	24.8 (17.4; 32.2) −7.0 *p* = 0.412	(2.37), *p* = 0.060
TC/HDL-c (mg/dL) ratio ∆pre-post (%) Pre-Post *p*-value	3.6 (3.1; 4.1) −1.1 *p* = 0.660	3.4 (3.0; 3.7) 1.1 *p* = 0.790	4.0 (3.4; 4.6) **−6.5** ***p* = 0.024**	3.7 (3.1; 4.4) −2.5 *p* = 0.419	(1.05), *p* = 0.384
MAP (mmHg) ∆pre-post (%) Pre-Post *p*-value	93.4 (88.6; 98.2) −1.3 *p* = 0.400	88.3 (83.1; 93.5) 2.7 *p* = 0.491	93.7 (88.8; 98.6) −3.0 *p* = 0.193	92.0 (82.5; 101.4) −0.02 *p* = 0.509	(0.88), *p* = 0.480
6 min walking test (m) ∆pre-post (%) Pre-Post *p*-value	630.1 (605.4; 654.9) **7.8** ***p* < 0.0001**	646.3 (619.1; 673.4) **10.9** ***p* < 0.0001**	647.3 (616.3; 678.4) **9.4** ***p* < 0.0001**	639.7 (598.7; 680.8) **6.5** ***p* = 0.037**	(0.28), *p* = 0.890

Data are shown as mean and 95% CI. HDL-c, high-density lipoprotein cholesterol; FPG, fasting plasma glucose; BMI, body mass index; TC, total cholesterol; LDL-c, low-density lipoprotein cholesterol; VLDL-c, very-low density lipoprotein cholesterol; TC/HDL-c, ratio total cholesterol/high-density lipoprotein cholesterol; MAP, mean arterial pressure; 6Mwt, six minute walking test. (∆pre-post) Delta percent % changes from pre- to post-intervention. Univariant test. Bold values denote significant pre-post changes at *p* < 0.05. baseline analyses carried out by Univariant ANOVA. *F*, denotes the Levene test. ^a^ denotes significant baseline differences vs. 20–29y group at *p* < 0.05. ^b^ denotes significant baseline differences vs. the 30–39y group at *p* < 0.05. ^c^ denotes significant baseline differences vs. 40–49y group at *p* < 0.05, ^d^ denotes significant baseline differences vs. 50–59y group at *p* < 0.05, using Sidak’s post hoc test.

**Table 2 jcm-10-05582-t002:** Correlations between the changes in the 6Mwt and MetS, anthropometric, and cardiometabolic outcomes.

Outcomes	*r* =	*p* Value
**Primary**		
∆6Mwt—∆WC	**−0.26**	***p* = 0.017**
∆6Mwt—∆HDL-c	−0.08	*p* = 0.460
∆6Mwt—∆FPG	0.03	*p* = 0.740
∆6Mwt—∆Tg	0.07	*p* = 0.499
∆6Mwt—∆SBP	0.12	*p* = 0.280
∆6Mwt—∆DBP	0.02	*p* = 0.801
**Secondary**		
∆6Mwt—∆BM	−0.21	*p* = 0.054
∆6Mwt—∆BMI	−0.07	*p* = 0.491
∆6Mwt—∆MAP	0.06	*p* = 0.575
∆6Mwt—∆TC	0.08	*p* = 0.481
∆6Mwt—∆LDL-c	0.00	*p* = 0.983
∆6Mwt—∆VLDL-c	0.07	*p* = 0.499
∆6Mwt—∆TC/HDL-c	−0.08	*p* = 0.460

6Mwt, six-minute walk test; WC, waist circumference; HDL-c, high-density lipoprotein cholesterol; FPG, fasting plasma glucose; Tg, triglycerides; SBP, systolic blood pressure; DBP, diastolic blood pressure; BM, body mass; BMI, body mass index; MAP, mean arterial pressure; TC, total cholesterol; LDL-c, low-density lipoprotein cholesterol; VLDL-c, very-low density lipoprotein cholesterol; TC/HDL-c, ratio total cholesterol/high-density lipoprotein cholesterol.

## Data Availability

The data presented in this study are available on request from the corresponding author. The data are not publicly available as they are part of a regional grant.

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
