# Peer review of "Similar Adaptations to 10 Weeks Concurrent Training on Metabolic Markers and Physical Performance in Young, Adult, and Older Adult Women"

_jcm, 2021, doi:10.3390/jcm10235582_

Round 1

Reviewer 1 Report

Dear authors, 

I really appreciated your modifications according to my suggestions.

Congratulations for this work.

Author Response

Dear reviewer, thanks for your positive comments. We have now also improved several other points that other reviewers reported.

Reviewer 2 Report

Many thanks for submitting this study for review and particularly on focusing on women. This work build on previous work done in the Chilean population, which has additional merit.  However, I have some significant concerns about the paper and study as it is currently presented.

Title: I think that this could be worded more clearly, as it doesn't read well/reflect the study design e.g. "young to older women" may suggest a repeated measures design rather than between groups.

Abstract: The background sentences are a bit vague and don't really represent the rationale/need for the study- consider revising.

Introduction: I would suggest you review this section in order to better represent the rational for the study.  At the moment, it is confusing as to what the main need/reason for it is i.e. you make societal health, physical activity and sedentary behaviour (not defined or differentiated) arguments and then go on to mention "interference effects"- which are distracting, and also don't attend to the matter of a female only study or the increased need for intervention with increasing age.  A clear statement about WHY there is a need for the study is required.

Methods:

  • More information about your inclusion and exclusion criterion in required- it's shown in the flow chart but not explained.
  • Why was FPG used as the power calculation variable? 
  • Where was WC measures taken, why and what was the inter-intra tester reliability of the tester?
  • Why use the 6min walk test? Provide a reference to support the method an its validity and reliability in this population.
  • Were the exercise sessions supervised? How long did they go in duration (relates to claims of time efficiency in discussion)?
  • How did you determine treatment fidelity- what was the minimum number of sessions that participants had to attend to be included? Why didn't you control for this?
  • Did you control for menstrual cycle or menopause status? Why? What effect might this have on your results?

Results:

  • What were the baseline data?
  • How did the MetS risk factors cluster at the beginning and end of the intervention?
  • Why not use minimal clinical difference as a statistical analysis- this may have provided more meaningful results considering your clinical focus and claims.
  • Why use BMI as an outcome variable, particularly as you've not clustered risk factors or mentioned classifications.

Discussion: I found much of this section to be a repeat of the results and summary of previous literature, rather than a critical discussion of the results. While you have identified some limitations, you have not mentioned or discussed controlling for other confounding variables such as diet, reproductive hormone fluctuations, treatment fidelity etc.

  • line 339 what does 'prime line' mean?
  • line 372 what does 'robust effects' mean?
  • lines 383-4 is this just relevant to older women, or ALL women at risk of MetS?
  • What is the actual clinical relevance of the study?
  • Is this method actually less onerous on participants?

Author Response

Many thanks for submitting this study for review and particularly on focusing on women. This work build on previous work done in the Chilean population, which has additional merit.  However, I have some significant concerns about the paper and study as it is currently presented.

Author Response: Dear reviewer, thanks for this constructive comment. Hope to collaborate in improving and clarify your concerns.

Title: I think that this could be worded more clearly, as it doesn't read well/reflect the study design e.g. "young to older women" may suggest a repeated measures design rather than between groups.

Author Response: Dear reviewer, thanks for the comment. Following this, we have changed the title as follows;

“…Similar adaptations to 10 weeks concurrent training on metabolic markers and physical performance between young to older adult women…”

Abstract: The background sentences are a bit vague and don't really represent the rationale/need for the study- consider revising.

Author Response: Dear reviewer, thanks for the comment. Following this, we have changed the background as follows;

“…Abstract: It has been proposed that the combination of high-intensity aerobic exercises and resistance training (RT), known as concurrent training (CT), improve metabolic syndrome (MetS) markers and that this exercise mixture in CT seems to dampen muscle anaerobic pathways in RT denominated “interference effect”. However, there is scarce evidence on its effects in women across different ages. Therefore, we sought to determine the effects of a 10-w…”

Introduction: I would suggest you review this section in order to better represent the rational for the study.  At the moment, it is confusing as to what the main need/reason for it is i.e. you make societal health, physical activity and sedentary behaviour (not defined or differentiated) arguments and then go on to mention "interference effects"- which are distracting, and also don't attend to the matter of a female only study or the increased need for intervention with increasing age.  A clear statement about WHY there is a need for the study is required.

Author Response: Dear reviewer, thanks for the comment. Following this, we have summarized more the “Introduction” section and have re-written more specific paragraphs about the main topic (Concurrent training, women, and age effects). Thus, we invite you to read our last “Introduction” version.

Methods:

  • More information about your inclusion and exclusion criterion in required- it's shown in the flow chart but not explained.

Author Response: dear reviewer, thanks by the comments. According with this, we have now included the criteria of inclusion/exclusion as follows;

“…The  eligibility criteria were as follows; i) to be women under physical inactivity [measured by the International Physical Activity Questionnaire (IPAQ)] applied previously in Chile <600 metabolic equivalents (MET)·min/week [1], ii) fasting hyperglycemia >100 mg/dL and <126 mg/dL [2], and to have additionally other MetS markers such as; iii) 1+ values above normal cutoffs for: body composition (i.e., WC >80 cm, that indicates “high cardiovascular risk” in South American population [3]); systolic (SBP) or diastolic (DBP) blood pressure of 130–139 mmHg and 85–90 mmHg, respectively to SBP/DBP, denoting ‘high blood pressure, or SBP and DBP >140 mmHg and 90 mmHg, respectively, denoting ‘hypertension’] [4]; or lipid profile (i.e., total cholesterol >200 mg/dL) [3]. Exclusion criteria were as follows; i) cardiovascular contraindications to exercise, ii) history of stroke, asthma, or chronic obstructive pulmonary disease, iii) muscle-skeletal disorders, ii) smoking, iii) living in extremely rural areas, iv) enrolled in other exercise training programs, or v) T2DM or HTN diagnosis. A compliance rate to the exercise program ≥70% was required for the participants in the intervention group to be included in the statistical analyses.…”

  • Why was FPG used as the power calculation variable? 

Author Response: Dear reviewer, thanks for the comment. Considering the metabolic syndrome outcomes of the sample (waist circumference, systolic and diastolic blood pressure, HDL-cholesterol, triglycerides, and fasting plasma glucose), we observed that our sample into these outcomes FPG is maybe one of the most sensitive outcomes, and where we do not detect baseline differences among groups. Following this, the SD changes in this outcome were minimum and confident to apply the power calculation of the sample size.

  • Where was WC measures taken, why and what was the inter-intra tester reliability of the tester?

Author Response: Dear reviewer, thanks for the comment. As we take the WC measurement with the same evaluator at pre and post-test, we decrease the error possibilities in this procedure. Thus, for example, we take always 2 or 3 measurements and calculate the mean, which finally is registered into our sheet of fieldwork. Out intra-class coefficient has been high to this outcome considering the 1st and 2nd measurement (ICC 0.952), which is reported as “higher” following some authors (the value of the reliability coefficient ranged from 0 to 1, where ICC<0 indicated “no reliability”, ≥0 but <0.2 “slight reliability”, 0.2 to <0.4 “fair reliability”, 0.4 to <0.6 “moderate reliability”, 0.6 to <0.8 “substantial reliability” and 1 “almost perfect reliability”. Geeta, A., Jamaiyah, H., Safiza, M. N., Khor, G. L., Kee, C. C., Ahmad, A. Z., ... & Faudzi, A. (2009). Reliability, technical error of measurements and validity of instruments for nutritional status assessment of adults in Malaysia. Singapore Medical Journal, 50(10), 1013.

  • Why use the 6min walk test? Provide a reference to support the method an its validity and reliability in this population.

Author Response: Dear reviewer, as we developed this research with a physically inactive population, we decided to use a walking test of low cardiovascular load, in which the 6 min walking test has no restrictions of intensity, and it’s self-imposed intensity, where under a previous professional guide the participants walk during 6 minutes looking for covering the major distance as possible, but always with the possibility to spot in any moment. Thus, following this point, we have now improved this section, adding more information as follows;

“…Participants were then instructed to walk as fast as possible according to with their individual capacity in a 100–meter track until completing the six minutes or before if participants manifested stopping. The total distance walked was recorded and registered in meters, and their validation and protocol have been widely described and used previously by the National Health Chilean Survey [5].…”

  • Were the exercise sessions supervised? How long did they go in duration (relates to claims of time efficiency in discussion)?

Author Response: Dear reviewer, thanks for this comment. Following these concerns, we have now included this information, as follows;

“…Concurrent exercise training program

Before the intervention, all subjects performed three sessions of familiarization of CT, and after this, the program was developed into 3 days/week (2 sessions guided by a professional, and 1 session self-guided, with previous education before starting the program). Each CT training session was divided into two parts: 1) resistance training (RT), where participants completed free weights RT exercises aimed at six major muscle groups (i.e., biceps curl; shoulder press; upper row; squat), and 2) high-intensity interval training (HIIT) in a cycle ergometer (BH®, model Carbon Bike Generator, Santiago, Chile). The resistance training modality consisted of three series of 1:1 minute work–to–rest ratios. On each series, the number of repetitions was maintained until fatigue, defined as reaching a rate of perceived exertion (RPE) of 8 on the 0–10 modified Borg scale [6]. The RT intensity corresponded to a 20–40% of one-repetition maximum test (1RM) for each exercise at the start of the study and progressed to a 25–50% of 1RM at the study end. The HIIT exercise modality included 60-second bouts of pedaling at 8–10/10 RPE, followed by 120 seconds of passive recovery on the ergometer, as previously described [7]. The number of bouts increased during the program starting from 3 between weeks 1 to 4 and progressing to 5–7 between weeks 5 to 7, and 8–10 between weeks 8–10. The total length of the CT session was in the mean of 54 min (RT; 24 min and HIIT; 12 to 30 min). The total number of CT sessions were of 30 (100% adherence= 30 sessions, where 20 sessions were supervised, and the other 10 were self-guided). The minimum adherence expected was of 70% of guided sessions (i.e., 14 sessions), and at least 50% adherence for the self-guided sessions (i.e., 5 sessions).…”

Additionally, we have now complemented the “Discussion” section of this concern of time-efficient, as follows;

“…Although it is difficult to compare our CT program (~54 min/session of RT and HIIT) with other more time-efficient regimes as HIIT that promote specific benefits at cardiovascular and metabolic health [8],, by contrast, comparing our CT with 1-year of endurance training [9]), it is relevant to highlight that 12-weeks of CT induce integrally relevant benefits at body mass, lipid profile, and endurance performance. Likewise, CT versus long-time endurance training appears as more ‘volume efficient for promoting more integral health benefits in physically inactive adults at risk of MetS. Thus, findings from the present study may have important health repercussions in favor of the prescription of CT to decrease the MetS risk factors in physically inactive adults.…”

  • How did you determine treatment fidelity- what was the minimum number of sessions that participants had to attend to be included? Why didn't you control this?

Author Response: Dear reviewer, as we recruited and studied physically inactive participants, and these were women, and considering that the exercise program was developed in centers located near their neighborhoods, we encourage weekly to all participants to maintain adherence to the program, particularly the self-guided day.

On the other hand, the minimum number of sessions to be included in the statistical analyses were included now into the manuscript as follows;

“…The number of bouts increased during the program starting from 3 between weeks 1 to 4 and progressing to 5–7 between weeks 5 to 7, and 8–10 between weeks 8–10. The total length of the CT session was in the mean of 54 min (RT; 24 min and HIIT; 12 to 30 min). The total number of CT sessions were of 30 (100% adherence= 30 sessions, where 20 sessions were supervised, and the other 10 were self-guided). The minimum adherence expected was of 70% of guided sessions (i.e., 14 sessions), and at least 50% adherence for the self-guided sessions (i.e., 5 sessions).…”

About the control of the adherence for sessions, it is also relevant to mention, that this study consisted of an exercise-training program for preventing the metabolic syndrome in women of different group-ages, but additionally where despite we do not include into the statistical analyses to those subjects that do not adhere to the criteria adherence, but however, any of the participants could be excluded from the program. Thus, the adherence was always stimulated but it was voluntary.

  • Did you control for menstrual cycle or menopause status? Why? What effect might this have on your results?

Author Response: Dear reviewer, thanks for this comment. Regarding this, we think that despite you are right in this concern; the fact of adding more control, as including one control group for example to the menstrual cycle outcome or menopause status might have more effectiveness, but at the same time could have increased the complexity of the study development. On the other hand, about menopausal status, it is relevant to mention that not at all of the individuals were at risk of symptoms of menopausal symptoms, as were those from the 40-40y and 50-59y groups. In this sense, for example, part of the control mechanisms that we internally follow, included giving a wide amount of information to participants about when? And under which symptoms of menopausal, or menstrual cycle to call for researcher’s team for advice exercise inconvenient. This information for urgent stop the adherence by exercise group also included known pain the first days of exercise, inflammation, bone pain, or other common symptoms associated with the sex condition that could ameliorate the exercise participation. In this line, it is relevant to mention that we do not receive urgent cell phones for any of these symptoms, and only daily comments at the first of the study were shared by patients, but did not include precisely menstrual or menopausal symptoms.

Following this, we added this factor as a limitation of our study, as follows;

“…interpretation and v) we do not control the menstrual cycle or menopausal symptoms by a control group, although we do not register reports of participants during the follow-up about this concern. S…”

Results:

  • What were the baseline data?

Author Response: Dear reviewer, baseline data is shown in Table 1, where is shown by primary and secondary outcomes. Additionally, as we do not focus the study on pre-post changes, only as an informative point for reads we added the delta pre-post changes, as well as each pvalue.

  • How did the MetS risk factors cluster at the beginning and end of the intervention?

Author Response: Dear reviewer, this information was solved and was added by a previous comment, as follows;

“…The  eligibility criteria were as follows; i) to be women under physical inactivity [measured by the International Physical Activity Questionnaire (IPAQ)] applied previously in Chile <600 metabolic equivalents (MET)·min/week [1], ii) fasting hyperglycemia >100 mg/dL and <126 mg/dL [2], and to have additionally other MetS markers such as; iii) 1+ values above normal cutoffs for: body composition (i.e., WC >80 cm, that indicates “high cardiovascular risk” in South American population [3]); systolic (SBP) or diastolic (DBP) blood pressure of 130–139 mmHg and 85–90 mmHg, respectively to SBP/DBP, denoting ‘high blood pressure, or SBP and DBP >140 mmHg and 90 mmHg, respectively, denoting ‘hypertension’] [4]; or lipid profile (i.e., total cholesterol >200 mg/dL) [3]. Exclusion criteria were as follows; i) cardiovascular contraindications to exercise, ii) history of stroke, asthma or chronic obstructive pulmonary disease, iii) muscle-skeletal disorders, ii) smoking, iii) living in extreme rural areas, iv) enrolled in other exercise training programs, or v) T2DM or HTN diagnosis. A compliance rate to the exercise program ≥70% was required for the participants in the intervention group to be included in the statistical analyses.…”

  • Why not use minimal clinical difference as a statistical analysis- this may have provided more meaningful results considering your clinical focus and claims.

Author Response: Dear reviewer, thanks for the comment. Although we could agree with your position, the pre-post training-induced changes, or the fact that exercise training could improve metabolic syndrome risk factors are not new. Then, our manuscript was focused on comparing age groups and testing about some potential or not differences in the exercise promoting benefits in MetS outcomes. Following this, we used scientific statistical by the 95% confident interval commonly used.

  • Why use BMI as an outcome variable, particularly as you've not clustered risk factors or mentioned classifications.

Author Response: Dear reviewer, thanks for the comment. However, we included BMI only as a secondary outcome to adding more information to future readers. When weight loss is present, and groups show obesity, the contrasting information is useful for readers.

Discussion: I found much of this section to be a repeat of the results and summary of previous literature, rather than a critical discussion of the results. While you have identified some limitations, you have not mentioned or discussed controlling for other confounding variables such as diet, reproductive hormone fluctuations, treatment fidelity etc.

Author Response: Dear reviewer, thanks for this comment. Regarding this, we have now, re-ordered a paragraph look to clarify the major critical discussion about agree-effects, as follows;

“…In the present study, we found that 10-week CT intervention, including both RT and HIIT, promoted positive training-induced effects on MetS markers, such as central adiposity, blood pressure, lipid profile, and fasting glucose in all age groups, which suggest that this type of training could be effective to improve MetS factors across age in women at risk of this syndrome. Importantly, we found that both primary and secondary outcomes could be positively, but differentiated, affected by CT intervention, which suggests that our results could be relevant for clinical interventions in this population. Thus, as our data revealed robust effects across all ages, it is relevant to summarize that the CT program, could promote beneficial effects independent of age. However, TC (Figure 3d) and LDL-c (Figure 3e) displayed a significant interaction between groups. Nevertheless, our pair comparison analyses did not reveal significant differences between all groups. Of note, the older 50-59y group were more sensitive for improving secondary outcomes as TC (∆-12.8 mg/dL) and LDL-c (∆-8.9 mg/dL) followed CT intervention, which suggests that this population could get more health benefits from CT; however, as our study did not detect significant differences across ages, future studies need to address this important point, increasing the number of participants and/or applying different CT strategies in this particular population.…”

On the other hand, following your comments about controlling for diet, reproductive hormone fluctuations, treatment fidelity etc.

Author Response: Dear reviewer, thanks for this comment. Regarding this, as the main topic is about “age-group” comparisons in the CT for improving MetS in women, It is difficult to increase the discussion in other concerns. Additionally, to add more control during a social study it is difficult to maintain the daily lifestyle, due to this was not a study under Lab conditions. Following this, we have now included these points al “limitations” in the appropriate section, as follows; 

“…do not register reports of participants during the follow-up about this concern, and vi) not controlled for reproductive hormone fluctuations, however, by our previous experience in physically inactive women groups, we do not report a reduction of beneficial effects. Some …”

  • line 339 what does 'prime line' mean?

Author Response: Thanks dear reviewer, this was a mistake of typing. Now, the sentence was replaced as follows;

“…Exercise is considered as a ‘first line intervention’ to prevent and …”

  • line 372 what does 'robust effects' mean?

Author Response: Dear reviewer, we denote that no interaction effects were found by age group. Thus, we have now replaced by the following sentences,

“…Although our data revealed not age-interaction across all ages in the MetS outcomes, we did not found differences between age groups, which suggests that CT program, could promote beneficial effects independent of age. However…”

  • lines 383-4 is this just relevant to older women, or ALL women at risk of MetS?

Author Response:  Dear reviewer, thanks for the comment. We have now deleted it, due to we consider that the sentence is not necessary.

  • What is the actual clinical relevance of the study?

Author Response: Dear reviewer, thanks for the comment. We have now added this information, as follows;

“…Our study also provides some clinical relevance, for example, a reduction in cardiometabolic risk factors can positively modify disease mortality, and healthcare expenditures [10]. Decreases of ~0.9 to 2.0 mmHg on SBP or DBP reduced major cardiovascular events by 10% in T2DM patients [11], and HDL-c increases (~10 mg/dL) can led other clinical beneficial implications. Thus, as we reported SBP decreases of 0.1 to -1.6 mmHg, and other HDL-c increases from +0.4 to +3.1 mg/dL, our study provide relevant clinical implications to these physically inactive cohort at risk of MetS.…”

  • Is this method actually less onerous on participants?

Author Response: Dear reviewer, although we tried to interpret this question, it was difficult to interpret. We found this comment a bit wide. Please clarify if this is needed.

  1. Seron, P.; Munoz, S.; Lanas, F. [Levels of physical activity in an urban population from Temuco, Chile]. Revista medica de Chile 2010, 138, 1232-1239, doi:/S0034-98872010001100004.
  2. ADA. 2. Classification and Diagnosis of Diabetes. . Diabetes Care 2017, 40, S11-S24, doi:10.2337/dc17-S005.
  3. Alberti, K.; Eckel, R.; Grundy, S.; Zimmet, P.; Cleeman, J.; Donato, K. Harmonizing the metabolic syndrome. A joint interim statement of the IDF Task Force on Epidemiology and Prevention; NHL and Blood Institute; AHA; WHF; IAS; and IA for the Study of Obesity. Circulation 2009, 120, 1640-1645.
  4. Mancia, G.; Fagard, R.; Narkiewicz, K.; Redon, J.; Zanchetti, A.; Böhm, M.; Christiaens, T.; Cifkova, R.; De Backer, G.; Dominiczak, A. 2013 ESH/ESC guidelines for the management of arterial hypertension: the Task Force for the Management of Arterial Hypertension of the European Society of Hypertension (ESH) and of the European Society of Cardiology (ESC). Blood pressure 2013, 22, 193-278.
  5. Vásquez-Gómez, J.; Castillo-Retamal, M.; Faundez-Casanova, C.; Carvalho, R.S.d.; Ramírez-Campillo, R.; Valdés-Badilla, P. Ecuación para predecir el consumo máximo de oxígeno a partir de la prueba de caminata de seis minutos en jóvenes sanos. Revista médica de Chile 2018, 146, 830-838.
  6. Gillen, J.B.; Percival, M.E.; Ludzki, A.; Tarnopolsky, M.A.; Gibala, M.J. Interval training in the fed or fasted state improves body composition and muscle oxidative capacity in overweight women. Obesity (Silver Spring) 2013, 21, 2249-2255, doi:10.1002/oby.20379.
  7. Álvarez, C.; Ramírez-Campillo, R.; Ramírez-Vélez, R.; Izquierdo, M. Effects and prevalence of nonresponders after 12 weeks of high-intensity interval or resistance training in women with insulin resistance: a randomized trial. Journal of Applied Physiology 2017, 122, 985-996.
  8. Gillen, J.B.; Gibala, M.J. Is high-intensity interval training a time-efficient exercise strategy to improve health and fitness? Applied physiology, nutrition, and metabolism 2014, 39, 409-412.
  9. Junior, L.C.H.; Pillay, J.D.; van Mechelen, W.; Verhagen, E. Meta-analyses of the effects of habitual running on indices of health in physically inactive adults. Sports medicine 2015, 45, 1455-1468.
  10. Zhang, P.; Zhang, X.; Brown, J.; Vistisen, D.; Sicree, R.; Shaw, J.; Nichols, G. Global healthcare expenditure on diabetes for 2010 and 2030. Diabetes research and clinical practice 2010, 87, 293-301.
  11. Turnbull, F.; Neal, B.; Algert, C.; Chalmers, J.; Chapman, N.; Cutler, J.; Woodward, M.; MacMahon, S. Effects of different blood pressure-lowering regimens on major cardiovascular events in individuals with and without diabetes mellitus: results of prospectively designed overviews of randomized trials. Archives of Internal Medicine 2005, 14, 1410-1419.